# Addressing the Molecular Mechanism of Longitudinal Lamin Assembly Using Chimeric Fusions

**DOI:** 10.3390/cells9071633

**Published:** 2020-07-07

**Authors:** Giel Stalmans, Anastasia V. Lilina, Pieter-Jan Vermeire, Jan Fiala, Petr Novák, Sergei V. Strelkov

**Affiliations:** 1Laboratory for Biocrystallography, KU Leuven, 3000 Leuven, Belgium; giel.stalmans@kuleuven.be (G.S.); anastasia.lilina@kuleuven.be (A.V.L.); pieterjan.vermeire@kuleuven.be (P.-J.V.); 2Department of Biochemistry, Charles University, 12800 Prague, Czech Republic; jan.fiala@biomed.cas.cz (J.F.); pnovak@biomed.cas.cz (P.N.); 3Institute of Microbiology of the Czech Academy of Sciences, 14220 Prague, Czech Republic

**Keywords:** nuclear lamins, intermediate filaments, X-ray crystallography, chemical cross-linking, mass spectrometry

## Abstract

The molecular architecture and assembly mechanism of intermediate filaments have been enigmatic for decades. Among those, lamin filaments are of particular interest due to their universal role in cell nucleus and numerous disease-related mutations. Filament assembly is driven by specific interactions of the elementary dimers, which consist of the central coiled-coil rod domain flanked by non-helical head and tail domains. We aimed to investigate the longitudinal ‘head-to-tail’ interaction of lamin dimers (the so-called A_CN_ interaction), which is crucial for filament assembly. To this end, we prepared a series of recombinant fragments of human lamin A centred around the N- and C-termini of the rod. The fragments were stabilized by fusions to heterologous capping motifs which provide for a correct formation of parallel, in-register coiled-coil dimers. As a result, we established crystal structures of two N-terminal fragments one of which highlights the propensity of the coiled-coil to open up, and one C-terminal rod fragment. Additional studies highlighted the capacity of such N- and C-terminal fragments to form specific complexes in solution, which were further characterized using chemical cross-linking. These data yielded a molecular model of the A_CN_ complex which features a 6.5 nm overlap of the rod ends.

## 1. Introduction

Lamins represent a distinct class within the intermediate filament (IF) protein family. These nuclear proteins are expressed in all human cell types. Lamin A (LA) and its splice variant lamin C as well as two closely related lamins B1 and B2 jointly form the lamina, a meshwork of ~3.5-nm-thick filaments located at the inner side of the nuclear envelope [1]. By doing this, lamins provide mechanical stability, which is the core function of IF family. Moreover, lamins are involved in a broad variety of cellular processes including chromatin organization and transcription, DNA replication and repair, cell differentiation, mitosis and gene expression [2,3,4,5,6,7]. As a result, mutations in the lamin genes cause a wide range of diseases called laminopathies, including Emery-Dreifuss muscular dystrophy and Hutchinson-Gilford progeria syndrome [8]. Furthermore, malfunctioning of lamins plays a role in diabetes, heat shock and cancer [9,10,11].

Like for the whole IF family, the basic constitutive unit of the lamin filament is a rod-like dimer which results from a parallel coiled-coil (CC) structure. In line with this, the primary structure of lamins reveals a central α-helical domain responsible for the CC formation which is flanked by non-helical N- and C-terminal domains called the head and the tail respectively. The central domain is divided into three CC segments known as coil1A, coil1B and coil2, which are interconnected by two linkers, L1 and L12 (Figure 1a). While the length of both coil1A and coil2 segments is almost universally conserved across the IF family, a signature feature of nuclear lamins is the coil1B segment, which is longer by 42 residues (six heptads) compared to cytoplasmic IF proteins [12]. The CC segments are characterized by a pattern of predominantly hydrophobic residues which mostly follow the classical heptad repeat resulting in a left-handed geometry. In addition, coil2 also contains several 11-residue (hendecad) repeats which yield a parallel (untwisted) α-helical bundle [13,14]. Recent crystallographic data revealed that both L1 and L12 linkers in lamins are α-helical even though the CC core is locally interrupted [12,15]. These observations confirmed an earlier suggestion that the linkers represent the points of flexibility within the rod. 

In stark contrast with the rod domain, the head and tail domains of IF proteins are dominated by regions predicted to be intrinsically disordered [16]. At the same time, both the head and the tail play an important role in the filament assembly by interacting with specific regions of the rod, and were suggested to accommodate a more ordered structure once the filament is formed [17,18,19]. In LA, the head domain is relatively short (26 residues) compared to cytoplasmic IF proteins. Specifically in lamins, the tail domain additionally contains a nuclear localization signal and an immunoglobulin-like globular domain (Figure 1a). 

Assembly of all IF types depends on specific interactions of the elementary dimers. However, the assembly pathway of nuclear lamins is distinct from that of cytoplasmic IFs. Early in vitro studies revealed that lamins can produce longer thin threads of dimers all oriented in the same way, suggesting a longitudinal (so-called A_CN_) interaction of dimers (Figure 1b) as a dominant feature [20,21,22]. Filamentous structures composed of two antiparallel, laterally associated long dimer threads (thus counting four lamin chains per cross-section) were also present. These results are in line with studies of in vivo assembled lamina using cryo-electron tomography that revealed 3.5-nm-thick filaments [23]. Recent X-ray crystallographic studies have established the molecular detail of the lateral interaction of half-staggered, antiparallel lamin dimers with coil1B segments aligned (so-called A_11_ interaction) [12,15], while additional half-staggered mode (A_22_) aligning coil2 segments was studied using chemical cross-linking [15]. Excessive lateral assembly in vitro, resulting in paracrystals, was also observed [21,24,25]. In general, the appearance of various in vitro assembled structures was dependent on environmental conditions like pH, ionic strength and presence of calcium [17,20,21,26,27]. 

In the past, crystallographic studies have been instrumental in developing a better understanding of the IF structure including lamins. While the full-length dimer is too elongated and flexible to be crystallized, shorter rod fragments could be crystallized and resolved [28]. This ‘divide-and-conquer’ approach has helped to unravel major parts of the rod domain in vimentin, keratins and lamins to atomic resolution. However, when using fragments, it is imperative to provide for the correct formation of parallel, in-register dimeric CCs. Indeed, shorter fragments may not oligomerize at all, form CCs with wrong multiplicity such as trimers, or antiparallel and staggered structures rather than parallel and unstaggered [29]. While also seen for numerous other CC proteins, such complications were documented for fragments of the IF rod domain. For instance, a short fragment of vimentin corresponding to isolated coil1A segment was initially crystallized as a monomer (PDB code 1GK7) and later found to be only marginally stable as a dimer in solution [30,31]. Another example is a C-terminal rod fragment of LA (residues 328–398, PDB code 2XV5) that was found to engage in an unexpected, staggered assembly [32].

To address these problems, stabilization of CC fragments through fusions to other domains was introduced in the past. The idea was that a specific capping motif (N- or C-terminal) would bring together the corresponding ends of the sequence of interest, thereby ‘bootstrapping’ the formation of the CC. A natural example here is the C-terminal motif, known as the ‘foldon’, of the trimeric CC protein fibritin from bacteriophage T4 [33]. An early attempt to stabilize the last 28 residues (385–412) of the rod domain of human vimentin involved an N-terminal fusion to the GCN4 leucine zipper [31]. The latter was chosen because it had been known as a prototype CC dimer with high thermal stability [34]. More recently, bacteriophage φ29 scaffolding protein Gp7 [35] and microtubule binding protein Eb1 [36] domains were used as N- and C-terminal caps, respectively. These capping motifs enabled X-ray structure determination of several myosin fragments as well as tropomyosin overlap [37,38,39].

In the current work, our aspiration was to address the molecular detail of longitudinal lamin assembly (Figure 1b) using X-ray crystallography as the main tool. In the past this question could be tackled by preparing shorter N- and C-terminal fragments of the rod domain (dubbed ‘mini-lamins’) and examining their interaction in solution [40]. Here we further refined this approach in two directions. First, the cloned N- and C-terminal rod fragments were supplemented with C-terminal and N-terminal capping motifs respectively (Figure 1a), to ensure correct dimer formation. Second, our fragments included adjacent regions of the head and tail domain respectively, since these regions were shown to be important for longitudinal lamin assembly [27,40,41].

As a result, we were able to firstly determine three crystal structures of individual LA fragments. These included two N-terminal fragments comprising residues 1 to 70 and 17 to 70 respectively, both C-terminally fused to the Eb1 domain, and a C-terminal fragment comprising residues 327 to 403, N-terminally fused to the Gp7 domain. The N-terminal lamin fragments in particular reveal a previously unknown β-structural motif located at the very N-terminus of the rod domain, and also provide evidence towards the CC unzipping within the coil1A segment. Second, we were able to study the interaction of N- and C-terminal rod fragments using gel filtration and chemical cross-linking. As a result, we propose a three-dimensional molecular model of the longitudinal A_CN_ interaction of nuclear lamins.

## 2. Materials and Methods

### 2.1. Cloning, Expression and Purification

The overall strategy towards cloning and recombinant protein purification was as described in [29]. Initially the DNA sequences for the capping motifs were purchased as gBlocks Gene Fragments from Integrated DNA Technologies (Leuven, Belgium). Prior to cloning, a Quick-Change site-directed mutagenesis was performed to introduce a F40C mutation into the Gp7 cap (see Appendix A for primers). Sequence and ligation-independent cloning was performed using a pETSUK2 vector [42]. The resulting plasmids encoded a 6xHis tag and a small ubiquitin-related modifier (SUMO) domain, followed by the sequence of interest.

Overexpression of obtained constructs (Appendix A) was done in *E. coli* Rosetta 2 (DE3) pLysS strain (Merck, Germany) by auto-induction in the ZYP-5052 medium [43,44]. Cells were harvested by centrifugation, resuspended in low-imidazole buffer (12.5 mM imidazole, 250 mM NaCl, 40 mM Tris HCl pH 7.5, 5 mM βME) containing a lysis mixture (10 mM MgCl_2_, 1% Triton X-100, 5% SIGMAFAST^TM^ inhibitor cocktail (Sigma-Aldrich, Overijse, Belgium), 1 mM PMSF and 100U Cryonase cold-active nuclease (Takara Bio Europe SAS, Saint-Germain-en-Laye, France)), sonicated and clarified by centrifugation. The supernatant was loaded onto a Ni-chelating column (His60 Ni Superflow resin, Takara Bio Europe SAS, Saint-Germain-en-Laye, France), pre-equilibrated with low-imidazole buffer. 6xHis-SUMO tagged chimeras were trapped on the column and eventually eluted by applying high-imidazole buffer (500 mM imidazole, 250 mM NaCl, 40 mM Tris HCl pH 7.5, 5 mM βME). 6xHis-SUMO tag was cleaved by overnight incubation (4 °C) with SUMO Hydrolase 7K (1:1000 ratio) while dialyzing against a low-imidazole buffer. Afterwards the cleaved mixtures were loaded again onto the Ni column. Chimeric fusions were eluted by low-imidazole buffer while 6xHis-SUMO tags were trapped on the column. Finally, size-exclusion chromatography was performed on a Superdex 200 Increase 10/300 GL column (GE Healthcare Europe, Diegem, Belgium) in 10 mM Tris HCl pH 7.5, 150 mM NaCl. Purified fractions were concentrated using Amicon^®^ Ultra filters with 3kDa cut-off (Merck Millipore, Overijse, Belgium). 

### 2.2. Crystallization and X-Ray Structure Determination

Purified individual proteins were screened for crystallization using commercial kits (Hampton Research, Molecular Dimensions, Qiagen and Rigaku). Subsequently, extensive optimization was performed using a Dragonfly robot (SPT Labtech, Hertfordshire, UK).

Gp7_F40C_-LA 327-403 (9 mg/mL) was crystallized at 4 °C by the hanging drop method using 35% (*v/v*) methanol, 0.2 M MgCl_2_ and 0.1 M HEPES (pH 8.2) as precipitant. Crystals were mounted on cryo-loops using mother liquor supplemented with 30% (*v/v*) glycerol. Native data were collected at beamline Proxima-1, Synchrotron Soleil (Saint-Aubin, France). Standard processing in XDS [45] was performed to obtain a complete diffraction data set up to 2.9 Å resolution. Towards phasing by anomalous scattering on sulphur atoms, an additional dataset was collected using the wavelength of 1.8 Å. Three 720° helical scans were performed on different spots of the same large single crystal. The data were merged using XSCALE [45], yielding a redundancy of 120 and a significant anomalous signal up to 3.2 Å resolution. 

The anomalous data were submitted to the Auto-Rickshaw pipeline using a web server [46]. Initial search for the positions of anomalous scatterers using SHELXD [47] was followed by processing in Phaser [48], including 12 sites per monomer. Experimentally phased map enabled the initial tracing of the structure, which could be further improved using Buccaneer [49]. Placement of the Gp7_F40C_ cap structure into the initial map helped to determine the right direction of the α-helices forming the CC. Eventually, the strongest anomalous scattering position was attributed to the disulphide bridge formed by the engineered cysteine residue within the Gp7_F40C_ cap. Further five scatterers corresponded to sulphurs present in residues Met278 (Met2 of the Gp7_F40C_ cap), Met345, Met349, Met352, and Met371 (Figure 2a). Of the remaining putative anomalous scatterers, two could be assigned to Ni ions. One of them is coordinated by residue His285. While no metals were present in the crystallization condition, Ni ions could have attached to the protein during purification procedure. 

LA 17-70-Eb1 (8 mg/mL) was crystallized at 4 °C by the sitting drop method using 35.7% (*w/v*) 1,6-hexanediol, 5% (*w/v*) PEG 1K and 0.1 M trisodium citrate dihydrate (pH 4.9). Crystals were mounted on cryo-loops using mother liquor supplemented with 30% (*v/v)* glycerol. Native data were collected at beamline I04, Diamond Light Source (Didcot, UK). Data processing to 1.83 Å resolution was performed with xia2 [50] using DIALS [51] for indexing, refinement and integration, POINTLESS [52] for space group verification and AIMLESS [53] for scaling. Xia2 suggested that usable data extended to 1.83 Å resolution. Initial phasing was performed via molecular replacement using the Eb1 cap (dimer) as search model in Molrep [54], which established the positions of two dimers per asymmetric unit.

LA 1-70-Eb1 (14 mg/mL) was crystallized at 19 °C by the sitting drop method using 0.1 M sodium phosphate monobasic monohydrate, 0.1 M potassium phosphate monobasic, 2.0 M NaCl and 0.1 M MES monohydrate (pH 6.5). Crystals were mounted on cryo-loops without any additional cryo-protectant. Native data to 2.83 Å resolution were collected at beamline P14, Petra III storage ring (Hamburg, Germany), and processed in the same way as for the LA 17-70-Eb1 construct. Phasing could be performed by molecular replacement using a part (starting with residue 35 and containing the full Eb1 cap) of the LA 17-70-Eb1 structure taken as a dimer.

For all structures, Coot [55] was used for interactive model rebuilding. Automated refinement was carried out using Refmac5 [56] and Buster [57]. CC geometry was analysed using TWISTER via an online server (https://pharm.kuleuven.be/apps/biocryst/twister.php) [58]. Pymol (Schrödinger, New York, NY, USA) was used to prepare figures. All statistics on crystallographic structure determination are shown in Appendix A.

### 2.3. Size Exclusion Chromatography (SEC) and Multi-Angle Light Scattering (MALS)

N- and C-terminal lamin chimeras were mixed in various molar ratios and incubated for 20 min at 4 °C. Thereafter the individual proteins and the mixtures (50 μL, 1 mg/mL total protein concentration) were run on a Superdex 200 Increase 10/300 GL at 20 °C using an Akta Purifier 10 system (GE Healthcare Europe, Diegem, Belgium). Analysis of the elution fractions was done through sodium dodecyl sulphate polyacrylamide gel electrophoresis (SDS-PAGE) followed by Coomassie staining overnight [59]. Molecular mass determination was performed using an inline SEC-MALS setup. Light scattering was detected using a Dawn Heleos (Wyatt, Santa Barbara, CA, USA) and results were analysed with ASTRA 5 software (Wyatt, Santa Barbara, USA). The experiments were repeated in two different buffers, 10 mM Hepes (pH 7.5) with 150 mM NaCl and 10 mM TEA (pH 7.0) with 150 mM NaCl, which corresponded to those used in different chemical cross-linking experiments (see next section). No difference in SEC elution profiles or MALS-derived molecular masses was observed.

### 2.4. Chemical Cross-Linking and Mass-Spectrometry (MS) Analysis

Cross-linking experiments were carried out according to Rozbesky et al. [60] with minor modifications. Initial optimization of the cross-linking procedure was through SDS-PAGE. Towards MS-based analysis, the major SEC elution peak corresponding to the complex of LA 17-70-Eb1 and Gp7_F40C-_LA 327-403 (0.4 mg/mL total protein concentration in 10 mM Hepes buffer (pH 7.5) with 150 mM NaCl) was subjected to cross-linking with disuccinimidyl dipropionic urea (DSPU, CF Plus Chemicals, Brno-Řečkovice, Czech Republic) in 1:50 molar excess. SEC-purified complex of LA 22-70-Eb1 and Gp7_F40C-_LA 327-403 (0.4 mg/mL in 10 mM TEA (pH 7.0) with 150 mM NaCl) was cross-linked with disuccinimidyl glutarate (DSG/d6-DSG, molar ratio 1:1, Creative Molecules, Victoria, Canada), disuccinimidyl dibutyric urea (DSBU, CF Plus Chemicals) and 1-ethyl-3-(3-dimethylaminopropyl)carbodiimide hydrochloride (EDC, Sigma-Aldrich, Saint Louis, MO, USA) in the molar excess of 1:75, 1:75 and 1:300, respectively. Prior to digestion, disulfide bonds were reduced with 20 mM TCEP at 56 °C for 20 min and alkylated with 20 mM IAA at room temperature for 20 min in the dark. Subsequently, protein samples were diluted five times with 50 mM 4-ethylmorpholine acetate (pH 8.5)/acetonitrile (90:10 *v/v*) and trypsin was added (1:20 ratio). Samples were digested overnight at 37 °C and reaction was quenched by adding trifluoroacetic acid to 0.1%.

The LC-MS/MS analysis was performed as described in [61]. More details are present in Appendix A. In all the spectra, chromatograms were deconvoluted using SNAP 2.0 algorithm integrated in Data Analysis 4.4 (Bruker Daltonics, Leipzig, Germany) and exported as mascot generic files (mgf). Mgf files were searched by StavroX 3.6.0.1 [62] or Merox 1.6.0.1 [63] with the following settings: cleavage at C-end of Lys, Arg and Tyr with a maximum of 5 missed proteolytic cleavages, fixed carbamidomethylation of cysteines and variable oxidation of methionines. Cross-linker specificity was set as follows: N-termini, Lys, Ser, Thr and Tyr for DSPU, DSBU and DSG; C-and N-termini, Lys, Glu and Asp for EDC. Error tolerance was set to 1.0 ppm for parent ions and 2.0 ppm for fragment spectra. All cross-linked positions (Appendix A) were manually checked.

### 2.5. Molecular Modelling

Prior to modelling, small modifications were made to crystal structures of both N- and C-terminal fusions:▪LA 17-70-Eb1 structure: a symmetric regular CC dimer without kinks was constructed by using chain B starting at residue 27.▪Gp7_F40C_-LA 327-403: the crystallographic model ending with residue 381 was extended by five residues which were ordered in the previously published LA coil2 structure (PDB code 1X8Y [64]).

Initial modelling of the heterotetrameric complex was performed manually in Pymol and Coot by opening up the N-terminal part of LA 17-70-Eb1 dimer and the C-terminal part of the Gp7_F40C_-LA 327-403 dimer and creating a tetrameric overlap. To this end, the structure of the GCN4 leucine zipper core mutant P-LI forming a tetramer (PDB code 1GCL) was used as a template. The length of the overlap was chosen such that:▪The heptad patterns of all four chains were in register just like in the homotetrameric template.▪The model satisfied as many distance restraints corresponding to the experimentally observed cross-links for the complex (Appendix A) as possible. 

Here, residue 27 of LA 17-70-Eb1 was used towards distance restraints for the missing residues in the flexible head region. 

After manual docking, the model was refined using the GalaxyRefineComplex tool [65] via the GalaxyWEB server [66] (http://galaxy.seoklab.org/). Details of the modelling process are provided in Appendix A.

## 3. Results

### 3.1. Design of LA Fragments

Four capped LA fragments were designed, overexpressed in recombinant *E. coli* culture and isolated to >95% purity (Figure 1a, Appendix A). Three N-terminal constructs incorporated residues 1–70, 17–70 and 22–70 of human LA respectively, fused to residues 215–251 of the Eb1 protein (C-terminal cap) [36]. These constructs correspond to the complete or truncated LA head domain followed by the complete coil1A segment (predicted to start with residue 27 and end with residue 67 [14]) and the first three residues of linker L1. In all cases, the fusion was performed in such a way that the heptad repeat pattern was preserved from the lamin part into the capping motif. In addition, a construct comprising residues 1–49 of the Gp7 protein (N-terminal cap) [35] fused by LA residues 327–403 was prepared. Here, the lamin part corresponded to the last 54 residues of coil2 (up to residue 380) followed by a small tail portion. Attachment of the Gp7 cap was done to preserve the hydrophobic pattern of coil2 which includes regular heptads interrupted by a single stutter insert at residues 327–330 (LARE) [64]. The last chimera additionally included a F40C point mutation, located at heptad repeat position ‘d’ within the Gp7 cap. This mutation was made to introduce a disulphide bridge in order to facilitate experimental crystallographic phasing, since the initial location of a disulphide from anomalous diffraction data is substantially easier than of isolated sulphurs [67]. After protein expression and purification, the disulphide bridge could be confirmed by a non-reducing SDS-PAGE (data not shown). 

### 3.2. Crystal Structure of Gp7_F40C_-LA 327-403

The crystal structure of the Gp7_F40C_-LA 327-403 fusion construct (Figure 2a) was established to 2.9 Å resolution. To this end, experimental phasing on sulphur atoms could be used, which had been facilitated by an engineered disulphide bridge in the dimeric Gp7_F40C_ cap. Crystallographic data reveal good electron density for the capping motif and the lamin part up to residue 381, i.e., exactly the predicted end of the rod domain [14]. The tail domain part (residues 382–403) is disordered in the crystals. Superposition of the Gp7_F40C_-LA 327-403 structure with the previously determined structure of the LA 305-387 fragment (PDB code 1X8Y [64]) (Figure 2b) shows high structural similarity and gives a root mean square deviation (RMSD) of 1.29 Å for 54 Cα-positions (residues 327–380). Interestingly, differences between the two structures are the most noticeable near the N-terminal end of the LA sequence, which corresponds to the stutter (residues 327–330). While the LA 305-387 structure becomes increasingly disordered towards its N-terminus, our fusion with the Gp7_F40C_ cap provides for a proper stabilization of the LA sequence. As a result, the CC geometry of Gp7_F40C_-LA 327-403 (Appendix A) reveals an unwinding that is in line with theoretical expectations for a stutter [58,68].

### 3.3. Crystal Structure of LA 17-70-Eb1 Fragment

Crystallographic data for the LA 17-70-Eb1 fusion construct were phased by molecular replacement, relying on the dimeric Eb1 cap as a search model. The crystal structure was refined to 1.83 Å resolution. The asymmetric unit of the crystal contains two copies of the dimer that are readily superimposable (Figure 3a, Appendix A). In line with the design, a continuous CC with regular left-handed geometry is formed all along the length of the chimera. Importantly, the structure gives a glimpse of the interface between the head domain and coil1A, as reliable electron density starting with residue 23 is seen in all four chains. In addition, chain A is ordered, already beginning with residue 18, which is due to a crystal contact with a symmetry-related molecule mediated by residue Leu21. The first α-helical residue is Arg28, which is in line with earlier predictions [14,15]. Interestingly, the side chain of Thr27 stabilizes the N-terminus of the α-helix by making a hydrogen bond with the main-chain nitrogen of Gln30 (Figure 3b). Such helical capping function of threonine and serine residues has been described before [69].

The N-terminal region of the dimer reveals two unexpected features. First, residues 23 through 27 corresponding to the proximal residues of the head domain form a β-strand, facilitated by a Pro residue in position 23. Two such strands form a short antiparallel β-sheet, further referred to as the ‘β-lock’, which flanks the N-terminus of the CC (Figure 3b). The residues Ile26 in the middle of the β-lock are symmetrically interlinked by two main-chain hydrogen bonds, while their hydrophobic side chains are pointing towards the CC. 

The second unexpected feature is that, consistently for both dimers in the asymmetric unit, one of the two chains reveals a sharp ~50° change in the α-helical axis direction (‘kink’) at residue Leu35. Main-chain dihedral angles for this residue are φ = −95°, ψ = 0. While still in the allowed region of the Ramachandran plot, these values are distinct from the typical values (φ = −65°, ψ = −45°) seen for the rest of the α-helix. This kink results in a loss of two standard i->i+4 main-chain H-bonds (between residues Asp34 and Leu38 and between Glu33 and Glu37, respectively). However, either bond is replaced by a H-bond ‘bridge’ accomplished by an ordered water molecule (Figure 3c). In addition, the standard H-bond between the main-chain oxygen of Lys32 and the nitrogen of Gln36 is in a suboptimal angular position.

Interestingly, the formation of the β-lock and the kink appear to be interdependent. Indeed, the latter causes a drastic deviation of the N-terminus of coil1A from the otherwise typical left-handed supercoiling (Figure 3a). In the absence of the kink the two strands made by residues 23–27 would have been located away from each other (Appendix A).

Coil1A of LA has a pronounced heptad pattern with predominantly hydrophobic residues in ‘a’ and ‘d’ positions for the most of its length. However, both the first residue of coil1A (Arg28) corresponding to an ‘a’ position and the following ‘d’ residue (Glu31) are not hydrophobic. Instead, polar interactions involving residues Arg28 and Glu31 of one chain and residue Asp34 of another chain (as well as symmetric interactions on the other side of the CC) are observed. The first residue actually creating the hydrophobic core of the dimer is Leu35 (Figure 3a). Our structure superimposes well with the recently published structure of a longer lamin fragment 1–300 (PDB code 6JLB) [15], resulting in a Cα RMSD of 0.94 Å for residues 28–70 (Appendix A). This latter structure consistently features Arg28 as the first α-helical residue and Leu35 as the first residue forming the core of coil1A, while not resolving any residues of the head domain.

### 3.4. Crystal Structure of LA 1-70- Eb1

In addition, we determined the crystal structure of the LA 1-70-Eb1 construct at 2.83 Å resolution. One dimer per asymmetric unit of the crystals is observed. Here the head domain is almost entirely disordered, as reliable electron density only starts from residue 25 (Figure 4). The structure shows a remarkable opening-up of the N-terminal part of the dimer extending up to residue 45. Upon application of the crystal symmetry, two ‘unzipped’ coil1A dimers form a 3-nm-long antiparallel N-terminal overlap (Figure 4). The overlap features a common hydrophobic core formed by residues Ile26, Leu38, Leu42, Tyr45, Ile46 and Val49 and a symmetric pair of salt bridges (Glu37|Arg41 and Arg41|Glu37) at each side of the overlap. The conformation and interactions of the N-terminal part of the dimer in the context of fusions LA 1-70-Eb1 and LA 17-70-Eb1 are thus entirely different. While in LA 17-70-Eb1 the entire coil1A forms a regular CC structure, which is flanked by the β-lock at the N-terminal end, in LA 1-70-Eb1 all these N-terminal interactions are lost and replaced by the antiparallel contact observed.

The large differences observed in the dimer structure and crystal lattice contacts between the LA 17-70-Eb1 and LA 1-70-Eb1 structures are most likely due to the different crystallization conditions, rather than a consequence of the complete head being retained in the latter case. Indeed, since the head domain in the LA 1-70-Eb1 structure is almost entirely disordered, it does not seem likely that it plays a major role in the formation of the antiparallel overlap of two coil1A dimers. At the same time, the observed structure clearly reveals the weakness and plasticity of the N-terminal part of coil1A. Indeed, unzipping of the CC has also been observed near the N-terminus of isolated coil1A fragment (residues 102–138) of human vimentin, even in the presence of a stabilizing mutation Y117L [30]. Sequence conservation between the coil1A regions of human LA and vimentin is rather high (70% similarity, see also Appendix A). In fact, the N-terminal opening-up of coil1A dimer is the most pronounced in the LA 1-70-Eb1 structure, followed by a smaller opening in the vimentin structure and a completely regular CC geometry in the 17-70-Eb1 construct (Figure 5).

### 3.5. Interaction of the N- and C-Terminal Chimeric Constructs in Solution

By design, our N- and C-terminal chimeras are very suited for examining the A_CN_ interaction of lamin dimers. As the first step, we studied their association using SEC. After Gp7_F40C_-LA 327-403 had been mixed in equimolar ratio with either LA 1-70-Eb1, LA 17-70-Eb1 or LA 22-70-Eb1, a single major chromatographic peak was revealed, eluting much earlier than any individual component (Figure 6a). Importantly, SDS-PAGE analysis of multiple chromatographic fractions across the peak revealed both individual proteins in a constant 1:1 ratio (Figure 6b). This pointed to the formation of a higher-order complex between the two constructs, presumably corresponding to the A_CN_ tetramer.

In addition, inline MALS analysis of the SEC elution peaks was performed. Molecular weights (MW) measured for all individual fragments i.e., LA 17-70-Eb1 (21.3 kDa), LA 22-70-Eb1 (19.6 kDa) and Gp7_F40C_-LA 327-403 (28.6 kDa) were in excellent agreement with the theoretical values for the dimers (21.2 kDa, 20.2 kDa, 29.7 kDa, respectively; Figure 6a). However, MW (value at the top of the peak) of the complexes was considerably lower than expected for a heterotetramer with 2:2 stoichiometry. For the LA 17-70-Eb1/Gp7_F40C_-LA 327-403 complex, the measured MW was 35.2 kDa, compared to a theoretical value of 50.9 kDa. For the LA 22-70-Eb1/Gp7_F40C_-LA 327-403 complex, the experimental MW was 32.4 kDa, compared to a theoretical value of 50 kDa. It should be noted that the elution peak of the complex was quite broad, while the measured MW value decreased considerably across the peak (Figure 6a). Another important observation is the persistent presence of a minor peak corresponding to the free N-terminal construct in the elution profile of the complex. This minor peak was seen in the SEC profiles of both LA 22-70-Eb1/Gp7_F40C_-LA 327-403 (after injecting an equimolar mixture of both constructs) and LA 17-70-Eb1/Gp7_F40C_-LA 327-403 (after injecting the major peak fraction from a previous SEC run; Figure 6a). 

These observations suggest that the interaction between the N- and C-terminal lamin constructs is relatively weak, meaning that there is a dynamic equilibrium between the complex and individual components, resulting in a broad elution peak. Of note, the complex eluted at a distinctly earlier position than each of the isolated dimers, despite the apparent average MW for the complex being only ~20% higher than that of the Gp7_F40C_-LA 327-403 dimer. Such elution indicates that the Stokes radius (and therefore at least one dimension) of the complex is considerably larger than that of each individual dimer.

The SEC-purified complexes of N- and C-terminal constructs were extensively screened for crystallization. Several crystal forms could be obtained. However, all crystals were found to be produced by the Gp7_F40C_-LA 327-403 fusion alone, as evident from SDS-PAGE analysis of the crystals. Moreover, preliminary X-ray characterization always yielded the same space group and cell parameters as seen for Gp7_F40C_-LA 327-403 alone. We hypothesize that the presence of crystallization agents was disrupting the complex, which we have shown to be relatively weak.

### 3.6. Chemical Cross-Linking

As the next step, both individual N- and C-terminal fusions as well as SEC-purified complexes thereof were subjected to chemical cross-linking using three monofunctional cross-linkers with primary reactivity against amino groups (DSPU, DSBU and DSG) as well as a heterobifunctional cross-linker EDC with reactivity against amino and carboxy groups. After optimization of the procedure, SDS-PAGE analysis revealed a range of cross-linked products in all samples. Specifically, for all cross-linkers, the bands corresponding to homodimers (at ~20 and ~30 kDa, respectively) were dominant for both the individual fusions and the complex (Figure 7). In addition, cross-linking the complex using EDC yielded a distinct band compatible with the ‘heterodimer’ (i.e., the result of cross-linking of one N-terminal and one C-terminal chain). This band was not apparent for the DSG or DSBU cross-linkers. Importantly, cross-linking of the complex with DSG or DSBU, but not EDC, resulted in a distinct band with a mass close to 50 kDa, which corresponds to a heterotetramer composed of two N-terminal and two C-terminal fragments, as well as two lower bands that could be interpreted as two types of heterotrimers. No higher bands than the heterotetramer were observed (Figure 7), suggesting that the latter is the largest species present in the sample in a prominent concentration.

Next, the complexes of N- and C-terminal fragments were cross-linked and subjected to LC-MS/MS analysis. As the result of experiments involving four different cross-linkers (DSG, DSBU, DSPU and EDC) a total of 51 cross-links were identified (Appendix A). Of those, seven involved the tail portion (not resolved in the crystal structure) of the Gp7_F40C_-LA 327-403 fusion. The remaining 44 cross-links were used in subsequent structural modelling.

### 3.7. Molecular Modelling of the A_CN_ Complex

Results of chemical cross-linking were supportive of a heterotetramer composed of two N-terminal fragments and two C-terminal fragments, i.e., the species representing the A_CN_ interaction between LA dimers. Overall, this complex should be composed of both the N- and C-terminal dimers in a parallel orientation and with some overlap length, for which several different values were reported in the literature (see Discussion). As a starting hypothesis towards the detailed molecular architecture of the complex, we took a heterotetrameric CC structure, an architecture that has been proposed previously [40,64]. To this end, the crystal structures of the LA 17-70-Eb1 and Gp7_F40C_-LA 327-403 dimers were manually docked together upon opening up the interacting ends of each dimer. The overlapping part was modelled using a parallel homotetrameric heptad-based CC structure as a template. The length of the overlap was varied to satisfy the maximal number of observed chemical cross-links (Appendix A, Figure 8a,b). 

Our optimized A_CN_ tetramer model (Figure 8) has an overlap of 6.5 nm, measured as the distance between the ends of LA rod (i.e., residue 27 within the N-terminal fragment and residue 381 within the C-terminal fragment), projected on the long axis of the heterotetramer. The overlap is thus longer than the length of coil1A alone (4.7 nm). The heterotetramer has a common hydrophobic core involving residues in ‘a’ and ‘d’ positions of both N- and C-terminal fragments (Figure 8c).

Out of a total of 44 cross-links used, 31 are compatible with the current A_CN_ heterotetramer model, given the maximal allowed distances between the respective Cα positions (Appendix A). These include, first, 21 cross-links with both ends corresponding to the same sequence (i.e., either intrachain or intradimer cross-links) and, second, 10 interdimer cross-links (Figure 8b). The latter group includes two EDC-based cross-links between residues Lys76 and Glu381/Glu383 and Glu65 and Lys378, respectively. These zero-length cross-links are particularly useful towards modelling as they provide the most stringent restraint, i.e., a maximum of 15 Å between Cα positions. In our current model, both cross-links correspond to a Cα-Cα distance of 12 Å. Of note, the Glu65-Lys378 cross-link was recently reported for the full-length LA [70].

At the same time, the remaining 13 cross-links (30%) could not be accounted for by the constructed A_CN_ model. Indeed, chemical cross-linking is known to produce artefacts, including linkages resulting from random collisions [71]. In our case, these 13 cross-links could be readily explained by an antiparallel association of two heterotetramers (Appendix A). While both our SEC experiments (Figure 6) and SDS-PAGE analysis of cross-linking products (Figure 7) did not explicitly reveal such octameric species, their presence in small concentration cannot be excluded.

The final A_CN_ model was additionally analysed using the program PISA (http://www.ebi.ac.uk/pdbe/prot_int/pistart.html, [72]), which suggested that the obtained assembly is stable in solution (Appendix A).

## 4. Discussion

Here we showed that carefully chosen short terminal fragments of lamin rod domain (following the concept of ‘mini-lamins’ originally proposed in Stuurman et al. [24]) can be conveniently used for studying the longitudinal assembly in molecular detail. A key element of our strategy was the use of relatively short (~50 residues) N- and C-terminal rod fragments that are additionally stabilized by capping motifs (Figure 1). Specifically, the Eb1 capping motif was attached at the C-end of several LA fragments that included a part of the head domain followed by the coil1A segment, and the Gp7_F40C_ capping motif was attached at the N-terminal end of a LA fragment corresponding to the last few heptads of coil2 and the beginning of the tail domain (Appendix A). Such fusions expose the opposite ends of the rod domain in the same way as they would be present in the context of a full-length lamin dimer. Correspondingly, the association of these two fusions in solution should reproduce the A_CN_ interaction between the full-length lamin dimers which is responsible for longitudinal assembly.

As demonstrated here, such fusions provide several benefits. First, the capping motifs used (and especially the Gp7_F40C_ cap that had been additionally stabilized by a disulphide bridge) provide for the correct assembly of dimeric, registered CCs even for short lamin fragments. Second, the capping motifs help to phase the crystallographic data. Indeed, both the Eb1 and the Gp7_F40C_ caps employed here consist of an α-helical region yielding a stable dimeric CC and an additional α-helix folding back onto it. In contrast to a bare CC segment, such cap structure is more suitable as a molecular replacement model, since it does not suffer from ‘internal’ symmetry (i.e., it does not overlap with itself upon a seven-residue register shift [29]). Following an alternative possibility, an engineered mutation to cysteine in the Gp7_F40C_ cap has facilitated experimental phasing using the anomalous scattering of sulphurs. Finally, in all three solved crystal structures, the capping motifs were involved in lattice contacts (as exemplified in Appendix A for Gp7_F40C_-LA 327-403 and in Appendix A for LA 17-70-Eb1). These observations suggest a likely beneficial role of including the capping motifs towards crystallization.

The three solved crystal structures contribute to a better understanding of the molecular organization of lamin dimer. In particular, the LA 17-70-Eb1 structure (Figure 3) gives a glimpse of the proximal half of the lamin head domain. Indeed, for two out of four chains in the asymmetric unit, the head part was found to be ordered. Although this was only a result of interactions with symmetry-related dimers in the crystal lattice, this observation supports an earlier hypothesis that the head domains of IF dimers, while being dynamic in isolation, become more ordered upon binding to other dimers [73]. This interaction appears to catalyse the filament assembly. Moreover, our crystal data indicate that the last five residues of lamin head domain can form a β-strand (residues 23–27). Two such strands yield a ‘β-lock’ flanking the N-terminus of coil1A, although this requires a formation of an α-helical kink at residue 35 in one of the chains of the dimer. Finally, the N-terminus of the α-helix was stabilized by a hydrogen bond provided by the Thr27 residue, in line with a frequently observed α-helical capping mechanism [69].

Of note, in lamin A, B1 and B2, the primary structure near the border of the head and coil1A domains is highly conserved (Appendix A), suggesting that the β-lock could be a common feature in all lamins. However, lack of sequence conservation in other IF chain types (Appendix A) speaks against a similar β-structure being formed there. For vimentin but not for other IF types, an additional short α-helix (denoted as the ‘pre-coil’ domain) is predicted upstream of coil1A. Interestingly, both the pre-coil domain and coil1A in vimentin seem to be stabilized by N-terminal capping Thr/Ser residues, just like we observed in LA coil1A. 

Importantly, of the two N-terminal LA fragments, the 17-70-Eb1 structure represents a CC with regular geometry, stabilized by the ‘β-lock’. In contrast, the 1-70-Eb1 structure features an opened-up, ‘unzipped’ coil1A segment, even though the presence of the Eb1 cap keeps the two chains together at the C-terminus. The observed unzipping (Figure 4) is especially interesting with respect to the possible mechanism of longitudinal IF assembly. Recently, mutation Y45C in LA which is linked to Emery–Dreifuss muscular dystrophy has been shown to be detrimental to the A_CN_ interaction [15]. Such behaviour was attributed to stabilization of coil1A dimer due to mutation. Indeed, this residue is in a ‘d’ position where the small hydrophobic cysteine side chain is preferred over tyrosine. These authors observed partial formation of a disulphide bond by the mutated residue, although the negative effect of the mutation on A_CN_ interaction was also observed in reducing conditions. Parallel observations were made in the past for vimentin coil1A region which was demonstrated to be only marginally stable [30]. Moreover, point mutation Y117L in vimentin coil1A (i.e., in a position equivalent to Y45 in LA) resulted in a loss of longitudinal assembly, indicating a key role of coil1A unzipping in the latter process [30]. More recently another vimentin mutation Y400L near the C-terminus of the rod was shown to have the same effect, suggesting that also the end of coil2 should be unzipping towards longitudinal assembly [74].

As the next step, using SEC we could show that the pairs of N- and C-terminal LA fragments stabilized by fusions with capping motifs indeed interact in solution. Our observations are in line with earlier studies using longer lamin fragments without capping motifs [40]. In particular, we could confirm that the A_CN_ interaction appears to be relatively weak in solution. Indeed, our SEC-MALS data hinted towards a dynamic equilibrium between individual fragments and the complex (Figure 6).

Finally, we applied chemical cross-linking to explore the 3D architecture of the heterotetrameric A_CN_ complex. In line with the demonstrated role of N- and C-terminal unzipping of the rod domain in assembly, our starting hypothesis towards the three-dimensional modelling of the complex was a parallel CC heterotetramer (dimer of dimers), following a suggestion outlined in [64]. We adjusted the overlap of the two dimers to maximize the fit to the chemical cross-links obtained, which resulted in an overlap of 6.5 nm (Figure 8). Of note, different values for the A_CN_ overlap have been suggested in the past. Early electron microscopy data for in vitro assembled lamin threads visualized using rotary metal shadowing suggested a 1–3 nm overlap [24], while recent cryo-EM studies of natively assembled lamin filaments are more compatible with an overlap of 10 nm [23]. In each case the overlap was estimated based on the expected length of lamin rod (50–51 nm) and the observed periodic appearance of the globular tail domains along the filament. A recent study [15] suggested an A_CN_ overlap of 14 nm, as calculated on the basis of a chemical cross-link between LA 1-300 and LA 286-400 (R388C) fragments and the A_11_ overlap known from crystallographic studies [12,15]. Most recently, chemical cross-linking studies of full-length human LA have yielded the A_CN_ overlap value of 5–6 nm [70], which is in accordance with our result.

To our knowledge, our heterotetrameric model is the first attempt to approach the A_CN_ interaction in molecular detail. Our data highlight the specific regions that are critical for the correct longitudinal assembly of lamin filaments and indicate that this assembly depends on a delicate balance of molecular interactions. These observations explain the fact that lamin mutations (such as the ones discussed above) located in the N- and C-terminal regions of the rod can be detrimental to the assembly, ultimately leading to the disease phenotype. At the same time, the signature globular domain (residues 434–552) present within the lamin A tail is known to be the hotspot of interactions between lamina and its protein partners such as actin [75] and BAF1 (barrier-to-autointegration factor) which further mediates the interaction with LAP2α [5] and emerin [76]. Since the globular tail domain is connected to the rod by a flexible region (as confirmed by the recent cryoEM data [23]) one should expect that the interactions with partners are preserved also when individual lamin dimers are incorporated into the lamina. Altogether, over 50 lamin-associated proteins have been described but for a majority them the details of the interaction are still elusive [77].

In the future, further efforts towards a structure with true atomic precision, using either crystallography or cryo-EM, should be undertaken. Indeed, it is hoped that additional A_CN_ complexes, based on newly designed N- and C-terminal fragments and/or additionally stabilized complexes (e.g., through cross-linking) could be prone to crystallization. At the same time, cryo-EM studies could employ longer constructs and/or full-length proteins.

## Figures and Tables

**Figure 1 cells-09-01633-f001:**
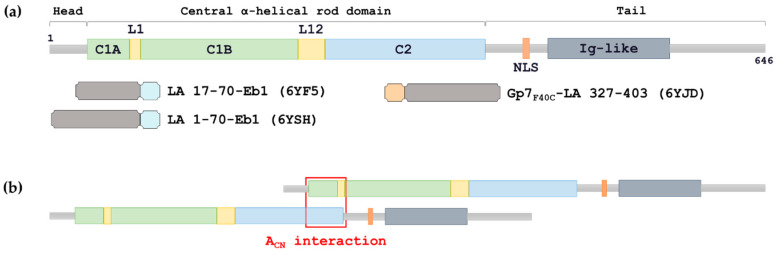
(**a**) Schematic representation of the primary structure of lamin A. C1A, coil1A; L1, linker 1; C1B, coil1B; L12, linker 2; C2, coil2; NLS, nuclear localization signal; Ig-like, immunoglobulin-like domain. Chimeric fragments used here for crystallographic studies are shown below. The capping motifs Eb1 and Gp7_F40C_ are coloured cyan and wheat, respectively. (**b**) Scheme of longitudinal lamin assembly which is based on the A_CN_ interaction.

**Figure 2 cells-09-01633-f002:**
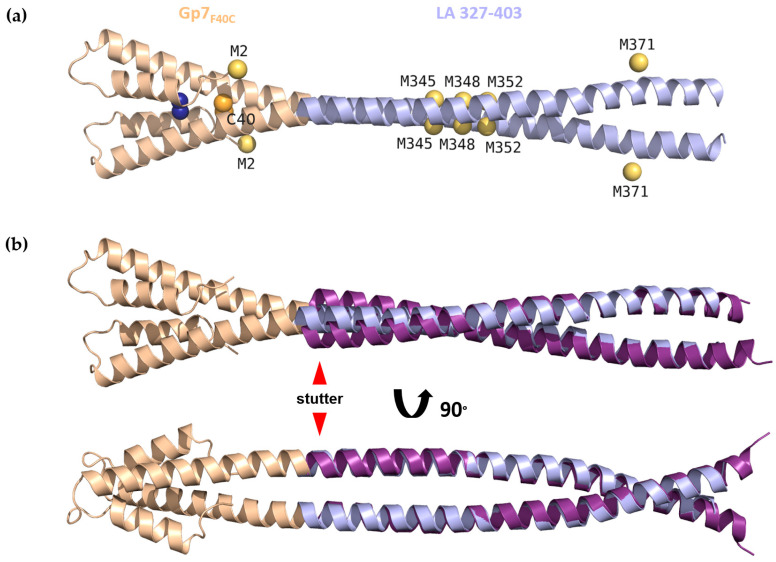
(**a**) Crystal structure of Gp7_F40C_-LA 327-403. The LA 327-403 region and Gp7_F40C_ cap are coloured light blue and wheat respectively. Sulphur atoms used for structure determination are coloured orange (cysteine) and yellow (methionine). Additional bound heavy atom ion (Ni) is coloured deep blue. (**b**) Superposition of the Gp7_F40C_-LA 327-403 structure with residues 327–386 of LA 305-387 structure (PDB code 1X8Y, purple) in two perpendicular views. The stutter (residues 327–330) is indicated by red arrows.

**Figure 3 cells-09-01633-f003:**
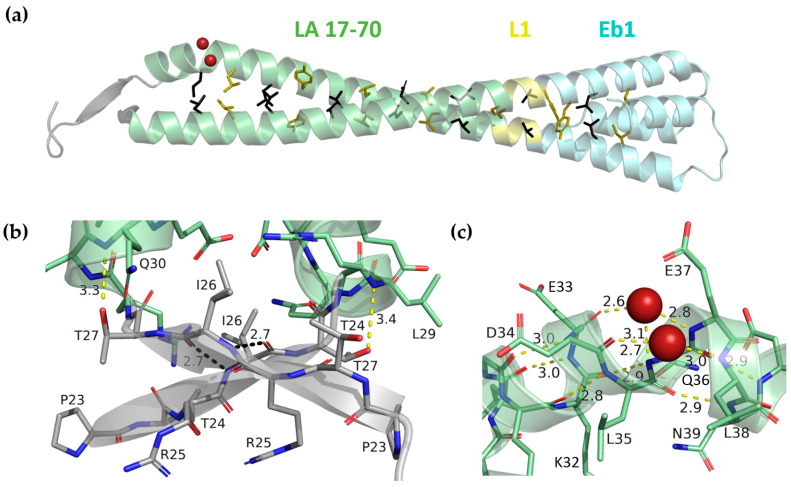
(**a**) Crystal structure of LA 17-70-Eb1. The head residues, coil1A and the first three residues of the linker L1 are coloured grey, pale green and yellow respectively. The Eb1 cap is coloured pale cyan. Two water molecules involved in H-bond bridges at the α-helical kink are shown as red spheres. Side chains of residues in ‘a’ (coloured black) and ‘d’ (coloured olive) positions starting with Leu35 i.e., the residues forming the hydrophobic core of the CC are shown as sticks. (**b**) Top view of the N-terminal end of the dimer featuring an antiparallel β-lock stabilized by two H-bonds (cyan dashed lines) between Ile26 residues. The H-bond between the side-chain oxygen of Thr27 and the main-chain nitrogen of Gln30 is shown (yellow dashed line). (**c**) Zoom in on the α-helical kink present in chain A. Regular main-chain hydrogen bonding pattern as well as two water-based H-bonding bridges are shown (yellow dashed lines). The suboptimal angular position of the H-bond between the main-chain oxygen of residue Lys32 and the nitrogen of residue Gln36 is shown as a purple dashed line.

**Figure 4 cells-09-01633-f004:**
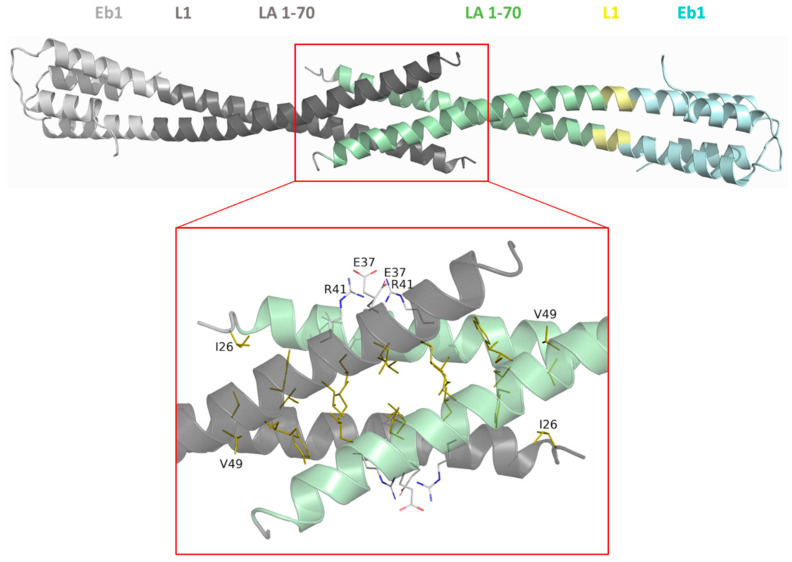
Crystal structure of LA 1-70-Eb1 dimer with coil1A in green, L1 region in yellow and the Eb1 cap in cyan. Additionally, a crystal symmetry mate is shown in black (lamin part) and grey (Eb1 cap). The zoomed area illustrates the interactions responsible for the antiparallel overlap of the N-terminal parts. Side chains forming the common hydrophobic core (coloured gold), as well as residues Glu37 and Arg41 forming salt bridges (coloured light grey) are shown.

**Figure 5 cells-09-01633-f005:**
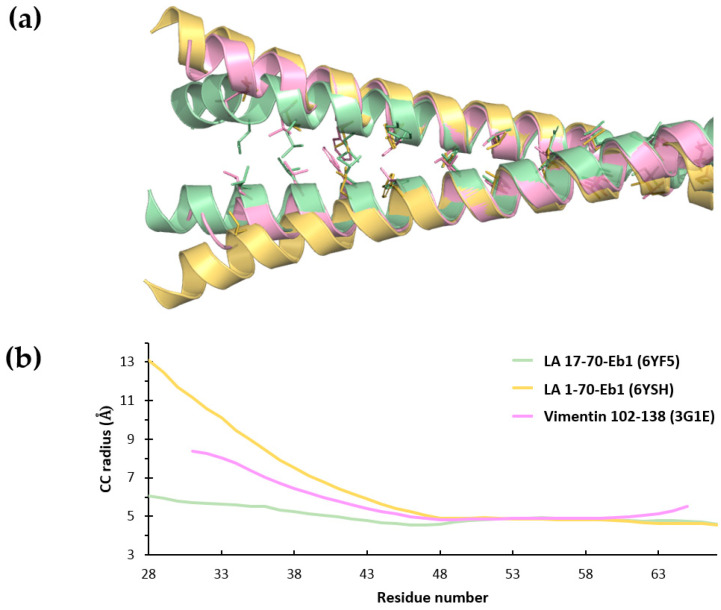
Analysis of the coil1A dimer opening in various structures (PDB codes are shown in brackets). (**a**) Superposition of the coil1A segments seen in LA 17-70-Eb1 (green), LA 1-70-Eb1 (gold) and vimentin 102-138 fragment with mutation Y117L (pink; PDB code 3G1E [30]). The residues in ‘a’ and ‘d’ positions forming the hydrophobic core are shown as sticks. (**b**) CC radius plotted as a function of residue number (according to the LA sequence) for the coil1A regions in the three structures, as determined using TWISTER [58].

**Figure 6 cells-09-01633-f006:**
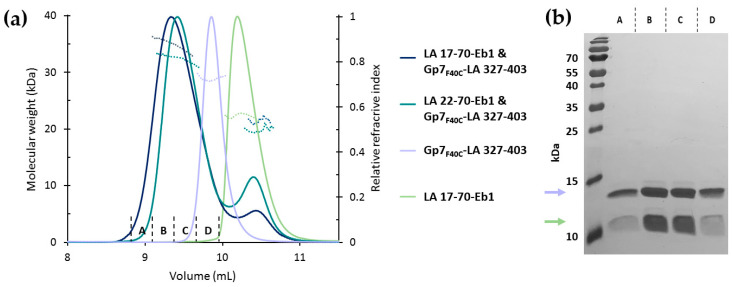
(**a**) SEC-MALS analysis of the LA 17-70-Eb1/Gp7_F40C_-LA 327-403 complex (1:1 ratio), LA 22-70-Eb1 and Gp7_F40C_-LA 327-403 complex (1:1 ratio), the Gp7_F40C_-LA 327-403 construct and the LA 17-70-Eb1 construct. The LA 17-70-Eb1/Gp7_F40C_-LA 327-403 complex and both individual constructs were run in 150 mM NaCl, 10 mM Hepes (pH 7.5) buffer. The second complex was run in 150 mM NaCl, 10 mM TEA (pH 7) buffer. Solid lines show the normalized refractive index profiles. Dashed lines show the MALS-based molecular weight values across the peaks. (**b**) SDS-PAGE of chromatographic fractions corresponding to the main peak of the LA 17-70-Eb1/Gp7_F40C_-LA 327-403 complex in (a). The blue and green arrows indicate Gp7_F40C-_LA 327-403 and LA 17-70-Eb1, respectively.

**Figure 7 cells-09-01633-f007:**
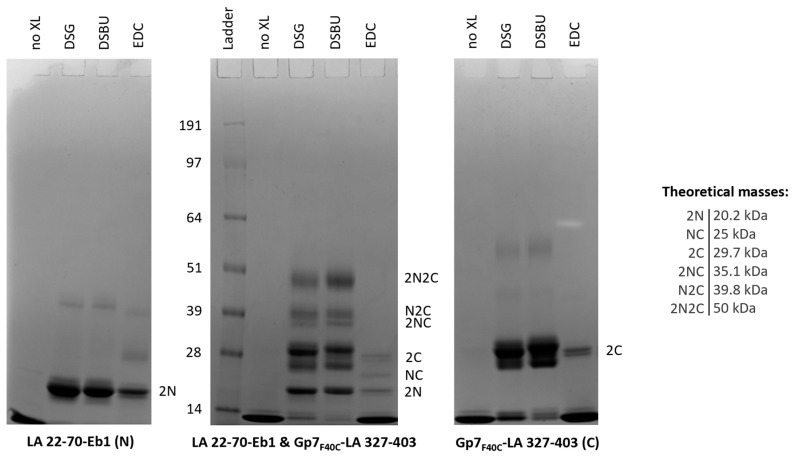
Reducing SDS-PAGE analysis of LA 22-70-Eb1, Gp7_F40C_-LA 327-403 and the SEC-purified complex of these two fragments without cross-linking (no XL) and after cross-linking with DSG, DSBU and EDC. The most likely stoichiometry of different bands is given at the right-hand side of each gel, where letters N and C indicate the N- and C-terminal fragments respectively. Theoretical masses of all bands are given in the side panel.

**Figure 8 cells-09-01633-f008:**
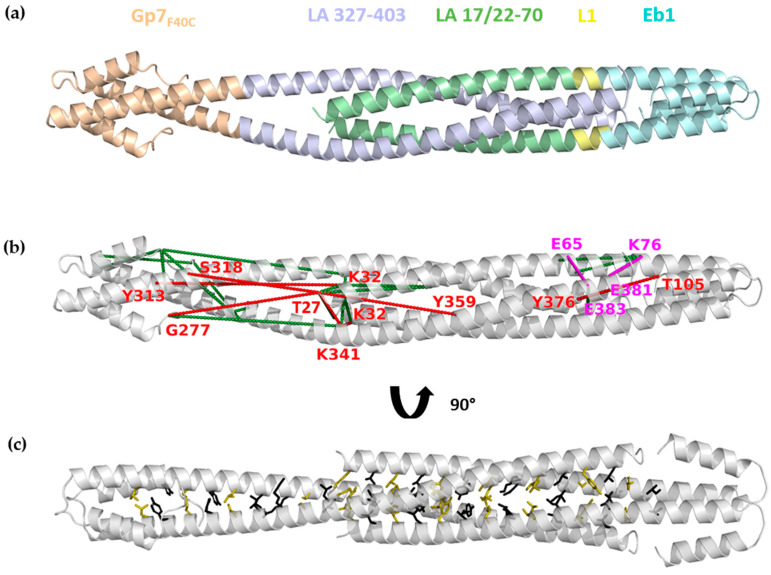
Modelling of the A_CN_ tetramer. (**a**) Final molecular model of the A_CN_ tetramer. Gp7_F40C_-LA 327-403 is coloured wheat (Gp7_F40C_) and blue (LA 327-403). LA 17-70-Eb1 is coloured green/yellow (LA 17-70) and cyan (Eb1). (**b**) Final molecular model of the A_CN_ tetramer with all 31 cross-links that were compatible with the model. Interdimer cross-links are coloured green. Tetramer cross-links are labelled and coloured red (DSBU, DSPU and DSG) and magenta (EDC). (**c**) Rotated view of the model with the hydrophobic core residues shown as sticks (‘a’ and ‘d’ positions in black and olive, respectively).

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
