# Peer review of "Addressing the Molecular Mechanism of Longitudinal Lamin Assembly Using Chimeric Fusions"

_cells, 2020, doi:10.3390/cells9071633_

Round 1

Reviewer 1 Report

The Manuscript by Stalmans G, et al., describes a series of experiments aimed to determine the molecular mechanisms behind the assembly of lamin A, using mainly an X-ray crystallographic analysis. Lamins are critical nuclear envelope proteins that modulates a variety of important cellular processes. Lamins are intermediate filaments with biochemical specific features that make them different from cytoplasmic intermediate filaments. Recent studies suggested that assembly of lamin A filaments used an antiparallel coiled-coil (CC) structure, where the basic unit is a rod-like dimer.

The authors employed a cleaver strategy based on the engineering of four recombinant fragments of human lamin A, mapping around the rod domain, because shorter fragments could be crystallized and resolved for crystallographic studies. They added capping motifs to the fragments for stabilization through fusions, to improve assembly. The ability of fragments to form specific complexes in solution was proven using SEC and cross-linking experiments. With the latter technique, authors obtained a band with a mass of ~50 kDa, which may correspond to a heterotetramer, composed of two N-terminal and two C-terminal fragments. They obtained crystal structures of two N-terminal fragments that fit well with previous reported structures, providing then a better understanding of the molecular organization of lamins.  Based on the hypothesis of a tetramer, the study finally provides a three-dimensional molecular model of the longitudinal interaction of fragments , with an overlap of 6.5 nm

In my opinion, this is an interesting and careful designed study that provide new clues for the assembly mechanism of lamin A. Structural assembly findings obtained could be expanded to other lamins. Furthermore, the strategy of using short capping short would facilitate in the future obtaining stabilized complexes prone to crystallization.  

The study deserves to be published in Cells after addressing the following point.

Major point

A brief discussion/statement of the potential implication of the results for the biology of the nuclear lamina. For instance, such assembly structure is compatible with multiple protein partners of lamin A? Or the possibility that some mutations causing nuclear envelopathies could affect the tetramer assembly structure.

Minor point

Some mistyping were found in the text, e .g. Introduction, lines 23. Please check it.

In Figure 2b, indicates stutter localization

In Supplementary Figure S4a, please put water molecules at the alpha-helical kink, as Figure 3a

In Supplementary Figure S4a, please indicate the chain name (AC BD) and the Leu35 with a sphere.

In Figure 3c, please indicate Lys32, Glu33 and Gln36 position.

Is there any report where the crystallization agent disrupts a complex?  

It appears that reference #70 is not mentioned in the main text, please check it.  

In Supplementary table S2, please include the number of hydrophobic residues and molecular weight of each construction.

Why different crystallization conditions were used between LA 1-70-Eb1 and LA 17-70-Eb1?

Reviewer 2 Report

Lamin A consists of the central coiled-coil rod domain flanked by non-helical head and tail domain. Stalmans et al., aimed to investigate the longitudinal ‘head-to-tail’ interactions of lamin dimers (ACN interaction) which is crucial for the filament assembly. They produced four recombinant fragments of human lamin A. These includes two N-terminal fragments comprising residues 1 to 70 and 17 to 70 respectively, both C-terminally fused to the Eb1 domain, and a C-terminal fragment comprising residues 327-403, N-terminally fused to the Gp7 domain. The authors determined crystal structure of three of them. They found a previously unknown beta structural motif located at the very N-terminus of the rod domain, and provided evidence towards the CC unzipping within the coil1A segment. Moreover, they studied the interaction between N- and C-terminal rod fragments and proposed a three-dimensional molecular model of the longitudinal ACN interaction. The results are novel and have impacts to the field. Therefore, I recommend this article to publish in Cells. However, there are several points to be addressed before publish.

Minor points;

  1. In the abstract, line 23, the authors mentioned “ano one C-terminal rod fragment”. Is this misspelling?
  2. They mentioned that “Four capped LA fragments were designed, overexpressed in recombinant E. coli culture…” line 253. However, only three of them are illustrated in Figure 1.
  3. “no XL” requires full spelling in the legend for Figure 7.

Author Response

Please see the attachment uploaded already, containing responses to both Reviewers
